# The Potential Clinical Use of Stem/Progenitor Cells and Organoids in Liver Diseases

**DOI:** 10.3390/cells11091410

**Published:** 2022-04-21

**Authors:** Christina Nikokiraki, Adriana Psaraki, Maria G. Roubelakis

**Affiliations:** 1Laboratory of Biology, Medical School, National and Kapodistrian University of Athens, 11527 Athens, Greece; noikokyr@biol.uoa.gr (C.N.); antrpsa@med.uoa.gr (A.P.); 2Centre of Basic Research, Biomedical Research Foundation of the Academy of Athens (BRFAA), 11527 Athens, Greece

**Keywords:** liver disease, clinical trials, organoids, EC, iPSCs, MSCs, hLPCs, cells

## Abstract

The liver represents the most important metabolic organ of the human body. It is evident that an imbalance of liver function can lead to several pathological conditions, known as liver failure. Orthotropic liver transplantation (OLT) is currently the most effective and established treatment for end-stage liver diseases and acute liver failure (ALF). Due to several limitations, stem-cell-based therapies are currently being developed as alternative solutions. Stem cells or progenitor cells derived from various sources have emerged as an alternative source of hepatic regeneration. Therefore, hematopoietic stem cells (HSCs), mesenchymal stromal cells (MSCs), endothelial progenitor cells (EPCs), embryonic stem cells (ESCs) and induced pluripotent stem cells (iPSCs) are also known to differentiate into hepatocyte-like cells (HPLCs) and liver progenitor cells (LPCs) that can be used in preclinical or clinical studies of liver disease. Furthermore, these cells have been shown to be effective in the development of liver organoids that can be used for disease modeling, drug testing and regenerative medicine. In this review, we aim to discuss the characteristics of stem-cell-based therapies for liver diseases and present the current status and future prospects of using HLCs, LPCs or liver organoids in clinical trials.

## 1. Introduction

The liver is the largest and most important metabolic organ of the human body and is responsible for several functions related to the maintenance of homeostasis. The liver is composed of several cell types, including hepatocytes, cholangiocytes, hepatic stellate cells, Kupffer cells and liver sinusoidal endothelial cells [1]. An imbalance of liver function can lead to several pathological conditions, known as liver failure, that are most commonly associated with increased inflammation, fibrosis, necrosis/apoptosis or even steatosis [2].

According to the World Health Organization (WHO), liver diseases are the 12th leading cause of mortality globally, with increasing incidence and prevalence. The causes can be related to genetic and/or drug-induced origins and can lead to acute or chronic liver disorders, such as hepatitis, fibrosis, cirrhosis, cancer and metabolic or autoimmune disorders [3]. Although the healthy liver exhibits an excellent regenerative potential, when it is subjected to a severe acute or chronic malfunction, this ability is limited [2]. Orthotropic liver transplantation (OLT) is currently the most effective treatment and the only established treatment for end-stage liver diseases and acute liver failure (ALF). However, its use is limited due to a shortage of available donors, long waiting lists, high costs and requirements for lifelong immunosuppression [4]. Therefore, alternative treatment options have been explored, including bioartificial livers, stem-cell-based therapies and hepatocyte or macrophage/immune cell transplantation [5].

Stem cells or progenitor cells derived from various sources are currently presented as an alternative source of hepatic regeneration, with a great importance placed on their ability to differentiate into hepatocyte progenitor-like cells (HPLCs) and hepatocyte-like cells (HLCs). Alongside these cells, hematopoietic stem cells (HSCs), mesenchymal stromal cells (MSCs), endothelial progenitor cells (EPCs), embryonic stem cells (ESCs) and induced pluripotent stem cells (iPSCs) are also known to differentiate into HPLCs and HLCs and can be used for the treatment of liver diseases in preclinical/clinical studies [6]. Among them, MSCs can undergo extensive proliferation in vitro, allowing them to quickly proliferate to the appropriate number for in vivo experimental approaches [7,8,9,10,11]. Other than that, MSCs produce a variety of growth factors that have been noticed to stimulate HPLC and HLC differentiation. HPLCs derived from MSCs can induce liver repair due to their paracrine effect and stimulate the downregulation of systemic inflammation in mouse models [9,11]. Moreover, the HLCs can effectively adopt a phenotype resembling hepatocytes, express hepatocyte-specific genes and perform glycogen storage and albumin synthesis functions [7,8]. More importantly, it has been recently observed that MSCs can release various factors and extracellular vehicles (EVs) that could effectively contribute to liver repair [10,12,13,14].

In addition, other cell sources of great importance in the field of therapy of liver diseases are ESCs and iPSCs. ESCs are derived from the inner cell mass of the blastocyst and can differentiate in vitro into hepatocytes, express liver-specific genes and mimic hepatic functions, although the risk of teratoma formation and tumorigenicity is relatively high [15]. On the other hand, iPSCs, which were firstly characterized in 2006 by Yamanaka et al., are derived from mature somatic cells and can be reprogrammed into an undifferentiated stage and exhibit the ability to differentiate into all cell types of the human body [16]. ESCs, iPSCs and even adult primary cells are known to be effectively utilized for the development of hepatic organoids. The formation of organoids can also be stimulated by co-culture with other cell types, such as MSCs or endothelial cells, in order to generate a three-dimensional (3D) structure [16]. These 3D structure organoids are also known as microtissues that can mimic in vivo conditions, such as cell-to-cell and cell–matrix interactions that involve the dynamic regulation of signaling pathways and paracrine signals [16]. Consequently, organoid technology represents a significant improvement in the 3D culture system, offering an enormous potential for use in preclinical applications and regenerative medicine.

In this review we attempt to summarize the characteristics of stem-cell-based therapies for liver diseases as well as to present the current status and future perspectives of using HLCs or liver progenitor cells (LPCs) in clinical trials.

## 2. Current Therapies for Liver Diseases

The gold standard therapy for liver diseases still remains the OLT, with a high survival rate of 88% for at least one year and 73% for five years. The liver transplantation can be performed as a whole organ transplantation or as partial liver transplantation (PLT) [17], giving the possibility of treatment of multiple patients from one donor. However, implications and limitations related to post-surgery thrombosis, biliary disorders and infections, graft rejection and high morbidity and mortality rates of patients [18] as well as long waiting lists, usually lead to extracorporeal support devices, necessitate new therapeutic strategies [19].

Therefore, current research strategies focus on novel therapeutic options for liver diseases, including artificial devices, cell therapy and organoids [20]. Regarding extracorporeal support for patients with liver diseases, several artificial devices have been developed to provide blood detoxification prior to liver transplantation. In particular, the molecular adsorbents recirculating system (MARS) and the fractionated plasma separation and adsorption (FPSA) system are both based on the membrane separation of toxins and are considered artificial liver support (ALS) devices for the treatment of patients with ALF [20,21,22]. Alongside these systems, bioartificial liver (BAL) devices have also been created as an alternative to artificial devices, using immortalized cell lines or xenografts to address liver metabolic functions [23] in the acute phase of liver diseases [22,23,24,25]. However, major problems, including the membrane weight cut-off of toxins, the risk of tumor formation [26] and zoonosis transmission [27], render their use in clinical trials debatable.

On the other hand, numerous clinical studies recently focused on different cell sources as potential treatments for liver therapies. In fact, the temporary efficacy of cell transplantation over OLT has been confirmed in several inborn disorders of liver metabolism [27,28,29,30]. Interestingly, hepatocytes have been reported as one of the most promising cell sources for the therapy of mild liver diseases, exhibiting numerous advantages compared to OLT. In particular, hepatocyte transplantation is considered to be a less invasive and expensive procedure, with the capacity of hepatocyte cryopreservation and direct use or re-use and with the benefit of treating multiple patients from one donor [28]. Nevertheless, many considerations, including the large-scale production, the donor shortage, the isolation technique and the culture and storage conditions as well as the cell morphology and function conservation after the transplantation [31,32,33,34,35], indicate the necessity of alternative cell sources. Following this direction, several clinical studies for liver diseases, that use fetal liver progenitor cells (FLPCs), human liver stem cells (hLSCs), HSCs, MSCs or HLCs, have been conducted. In more detail, FLPs are known to have the ability to engraft, proliferate, mature and differentiate into hepatocytes or cholangiocytes as well as to improve liver function in clinical application, whereas their use raises ethical issues. Furthermore, it has been reported that hLSCs successfully proliferate in vitro and differentiate into hepatocytes, with preclinical studies supporting their safety. Moreover, several clinical and animal studies have also introduced the importance of HSCs for the treatment of liver diseases, highlighting their contribution in hepatic regeneration; although the exact mechanism of their function still remains unclear, MSCs may also play an important immune-modulating role through their paracrine effects [7,12], with several clinical trials to be constantly registered worldwide [20]. Collectively, stem-cell-based therapies have also paved the way for the generation of functional HLCs, which are currently considered as a very promising clinical “tool” for liver malfunctions. Additionally, new bioengineered cell therapy solutions are currently being developed, such as liver organoid development, decellularized/cellularized liver scaffold generation, liver biofibrication and bioprinting in order to recapitulate liver tissue for clinical application [20].

Furthermore, liver organoids are currently being developed and referred to 3D cultures that allow the in vitro generation of tissue-like structures, mimicking the organization and the function of the fetal and adult tissues of organs. Organoids are considered to be an important therapeutic tool in toxicological, modeling and drug discovery studies because they form tissue-like structures [36], respond better to apoptotic drugs and correctly metabolize molecules [36,37,38]. However, the generation of organoids is a time consuming and demanding procedure, with limited clinical trials confirming their efficacy in liver malfunctions [39]. In more detail, the need of extra supportive biomaterials and growth factors, the culture conditions and the selection of an optimal culture device render the organoid therapeutic approach complicated enough [36]. In addition, decellularized/cellularized liver scaffolds that support the engraftment of liver cells as well as liver biofabrication and bioprinting with laser pulses and inkjets that mimic the liver structure represent strategies developed to support functionally active 3D liver tissues in vitro. However, their application in clinical studies remain controversial, mainly due to the lack of vascularization and the poor viability of organoids after transplantation as well as the mechanical and physical implications of culture function, [19,38,39,40,41] requiring the need of further investigation for their therapeutic effectiveness.

## 3. HLC Generation from MSCs

Hepatocytes play an important role in the metabolism of lipids, glucose, urea, bile and drugs as well as in cytochrome P450 and synthetic function. In the liver, hepatocytes can be generated from two different cell sources:LPCs, also known as oval cells in rodents, which are localized at the canal of Hering or at the intrahepatic bile ductules. They express hematopoietic cell markers, such as CD45 and CD109 [42] or CD34, and CD133 [43] and their efficacy has been tested in chronic and acute liver diseases [6,42,44];LSCs that exhibit similar differentiation, immunosuppressive and regenerative characteristics to MSCs, and are negative for the expression of hematopoietic cell markers [45].

The use of hLSCs has recently been reported as an alternative cell-based therapy in a clinical phase I study in infants with the genetic/metabolic disease inherited neonatal-onset hyperammonemia [46] as well as in a phase I/II clinical trial in pediatric urea cycle disorder and Crigler–Najjar syndrome [47]; however, their functional characteristics in liver regeneration remain unclear. Alongside these studies, hHLSC-EVs have also been investigated for their potential therapeutic outcome in regenerative, metabolic and cancer liver disorders due to their ability to transmit the desired protein, miRNA or mRNA to the target cells [48].

HLCs can be generated from three cell sources: (a) ESCs [49], (b) iPSCs [50,51] and (c) MSCs that can be sourced from various tissues, including the bone marrow (BM), umbilical cord (UC), blood, amniotic fluid (AF), scalp tissue, placenta and adipose tissue (AD) [52,53].

MSCs are multipotent cells, with several studies confirming their regenerative efficacy. Interestingly, MSC-based therapies present limited or no ethical issues and overcome the procedure of gene manipulation, making them a challenging therapeutic option for liver diseases. According to several publications, the differentiation process of MSCs into HLCs includes a procedure of induction and maturation [54] (Figure 1). The differentiation process to HLCs is also dependent on several differentiation factors, such as growth factors, cytokines and biochemicals. The induction step is conducted by the presence of EGF and FGF that stimulate the proliferation and the endodermal cell formation. The hepatic differentiation is normally induced by the administration of FGF, HGF, nicotinamide (NTA) and insulin transferrin selenium (ITS). In more detail, HGF is involved in the differentiation, chemotactic migration and proliferation of MSCs. In addition, oncostatin M (OSM) and dexamethasone (Dexa) are usually required for the maturation, in conjunction with FGF, ITS and HGF. Histone deacetylases such as trichostatin A (TSA), ascorbic acid and sodium butyrate have also been found to contribute to MSC differentiation into HLCs [55] (Figure 1). Additionally, several studies have confirmed the crucial role of miRNAs in HLC generation, including the study by Davoodian et al., who discovered the important role of miR122 during the differentiation of AD-MSCs into HLCs [56] (Figure 1).

However, HLCs exhibit limited function (estimated at 40% compared to primary hepatocytes), based on albumin (ALB) and alpha-1 antitrypsin (AAT) synthesis, urea production and cytochrome (CYP) activity [57,58]. Despite the fact that the 2D culture differentiation protocol of MSCs into HLCs appears to be effective in several studies of liver therapy, the 3D culture with the addition of HGF and FGF cytokines is considered to be beneficial for clinical applications. However, several parameters need to be tested further, including the protein secretion of hepatocyte markers (*ALB*, *AFP* and urea) and the enzymatic function (LDL uptake, glycogen storage and cytochrome P450) in order to validate the quality of the generated hepatocytes for clinical trials. Other important checkpoints of routine tests that also have to be taken into account after the HLC transplantation in clinical trials are the Model for End-Stage Liver Disease (MELD) score, which refers to bile clearance from the liver, the international normalized ratio (INR), which shows the ability of the liver to form coagulative factors, and the serum sodium levels [55].

## 4. MSC Use in Clinical Trials for Liver Diseases

Recently, cell-based therapies for liver diseases have focused on MSC transplantation due to their competitive advantages and important role in tissue repair and regeneration (Figure 2). As reported in the previous section, MSCs are multipotent progenitor cells that originate from the embryonic mesoderm and can be isolated from a variety of fetal and adult sources [59,60]. Moreover, fetal MSCs are capable of differentiating into multiple cell types, including hepatocytes, and release paracrine soluble factors and EVs that can contribute to tissue repair [12,19]. It is known that these cells can migrate themselves or promote other cells’ migration to sites of injury as a response to cellular damage signals [61]. Due to these beneficial characteristics, it has been shown that MSCs are able to prevent the progression of liver injury and improve liver function in vivo [62].

According to the NIH clinical trials database (https://clinicaltrials.gov/, 1 March 2022), 63 studies using MSCs as a treatment for liver diseases are registered. The majority of them are related to liver cirrhosis and appear to have completed clinical status. Interestingly, the most common cell types used in liver disease therapy are the BM-MSCs and/or UC-MSCs (Figure 2). More specifically, there are two studies using both allogenic BM-MSCs for Wilson’s disease (WD) (NCT01378182) and liver cirrhosis (LC) (NCT01591200). In particular, in 2021, Gupta et al. administrated stempeucel^®^ via the hepatic artery to LC patients in a phase II clinical trial. Stempeucel^®^ was the investigational drug containing cultured BM-MSCs at passage five [63]. According to their results, stempeucel^®^ injection was well tolerated, although there were no clinical improvements in any group and no differences between groups in the liver function test (LFT), MELD or Child Pugh scores [63]. In addition, there was a statistically significant improvement in the 2.5M (2.5 million cells/kg body weight) group compared to the 2.5M control group (without cells) in bodily pain, mental component summary, vitality and social functioning, whereas no improvement was reported between the 5M study groups [63].

Regarding autologous BM-MSC transplantation, 1 study is registered (NCT00956891) for HBV-related liver failure (HBV-LF), and 11 studies are registered for LC (NCT00993941, NCT00476060, NCT01454336, NCT01499459, NCT01741090, NCT01875081, NCT03838250, NCT04689152, NCT01724697 and NCT03209986), with 4 of them being related to alcohol (NCT01741090, NCT01875081, NCT03838250 and NCT04689152) and 2 of them being related to HBV (NCT01724697, and NCT03209986). In 2016, Suk et al., after hepatic injection of 5 × 10^7^ cells/mL, observed an improvement in fibrosis and in Child Pugh’s score in alcoholic patients [64]. According to these trials, autologous BM-MSCs therapy improved the hepatic fibrosis and liver function of patients with alcoholic LC. [24,25]. Moreover, in 2012, Kim et al. injected 5 × 10^6^ cells/mL of autologous BM-MSCs via the hepatic artery into patients with LC related to alcohol, which resulted in an improvement in their histology and Child Pugh’s score as a primary outcome [65]. A study by Kantarcioglu et al., estimated the impact of autologous BM-MSC transplantation in liver tissue and proposed that autologous transplantation of BM-MSCs is a safe and feasible procedure for LC patients [66]. More importantly, patients who received 1 × 10^6^ cells/kg via a peripheral vein exhibited an improvement in MELD score and increased levels of serum albumin [66]. As for patients with non-responder hepatitis C virus (HCV), the RNA levels of *HCV* were negative after the BM-MSC transplantation [66]. However, histopathological examination showed no significant changes in the liver tissue or liver fibrosis, and BM-MSCs did not engraft to the liver tissue in a sufficient way [66]. Furthermore, in 2011 Gholamrezanezhad et al., after intravenous injection of 400 × 10^6^ BM-MSCs into patients with LC, observed that the percentage of cells implanted to the liver was increased in the 10th day after the infusion [67]. Currently, there is one completed trial (NCT00956891) and two randomized controlled trials (NCT01724697 and NCT03209986) in the process of recruiting patients to determine the potential clinical benefits of MSC-based therapy for patients with HBV-LF and LC, respectively. In more detail, in 2011, Peng et al. transplanted MSCs into the liver via the hepatic artery with a 100% success rate, and, after 2–3 weeks, they observed improved levels of MELD score, alanine aminotransferases, albumin, total bilirubin and prothrombin time [68]. Based on this, autologous MSC transplantation is safe for patients with HBV, with a favorable short-term efficacy [68].

In addition, the transplantation of UC-MSCs as a treatment for liver diseases has also been investigated in 11 clinical trials (NCT01342250, NCT01220492, NCT05227846, NCT03945487, NCT05121870, NCT05224960, NCT04522869, NCT05155657, NCT01728727, NCT03826433 and NCT05106972) for LC, 1 trial (NCT01662973) for primary biliary cirrhosis (PBC), 1 trial (NCT04822922) for acute-on-chronic liver failure (ACLF) and 2 clinical trials (NCT01724398 and NCT01218464) for HBV-LF. In 2021, the study by Shi et al. proved that triple intravenous injection of 0.5 × 10^6^ UC-MSCs/kg at 4-week intervals resulted in a significantly higher overall survival rate, with improved liver function as determined by the levels of serum albumin, prothrombin activity, total bilirubin and cholinesterase, with no reported side effects [69]. In similar clinical trials, UC-MSCs have been used as a treatment for HBV-LF with (NCT01724398) or without (NCT01218464) plasma exchange (PE). In particular, in 2019, Xu et al. aimed to investigate the efficacy of the combined transplantation of UC-MSCs and PE as a treatment for HBV-related acute-on-chronic liver failure (HBV-ACLF) [70]. Their results showed that the levels of total bilirubin, alanine aminotransferase, aspartate transaminase and MELD score were significantly decreased during treatments; however, this clinical approach could not significantly improve the short-term prognosis of HBV-ACLF [70]. Currently, there are four clinical trials in the recruiting phase (NCT01662973, NCT04822922, NCT03826433 and NCT05106972) using UC-MSCs as a treatment for PBC, ACLF and HBV-related liver cirrhosis. In 2013, Wang et al. infused 0.5 × 10^6^ UC-MSCs/kg via a peripheral vein in patients with PBC who had an incomplete response to ursodeoxycholic acid (UDCA) therapy [71]. Their results exhibited a significant decrease in serum alkaline phosphatase and g-glutamyltransferase levels after 48 weeks, indicating this method as a novel therapeutic approach [71].

On the other hand, a combination of stem cells/progenitor cells was used in a number of clinical trials. For example, in a phase II clinical trial in 2022, Sharma et al. (NCT04243681) proposed that the combination of HSCs with MSCs, which has an immunomodulatory effect, may enhance the duration of cell therapy improvements on liver function [72]. In more detail, CD34^+^ HSCs and MSCs were infused through the hepatic artery, and 3 months later MELD score and serum albumin level improvements and the absence of hepatic encephalopathy or gastrointestinal bleeding were reported [72]. Moreover, a number of clinical studies utilized both BM-MSCs and UC-MSCs in LC (NCT01877759) and HBV-LF (NCT01844063) in order to evaluate the safety and efficacy of the cell treatment. The use of autologous AD-MSCs as a potential therapy of LC has also been explored in the NCT01062750, NCT02297867 and NCT03254758 clinical trials. Especially, the 2015 study by Lin et al. showed that intrahepatically injected autologous AD-MSCs are able to differentiate into HLCs in the damaged liver tissue and contribute to the repair of liver fibrosis [73]. The main findings of using MSCs as a therapeutic transplantation tool in liver diseases are summarized in Table 1.

## 5. HLC Use in Clinical Trials for Liver Diseases

Since hepatocytes have a crucial role in liver function, nowadays the clinical approach for liver diseases is focused on the transplantation of MSC-derived HLCs and LPCs. Currently, according to clinical trials database (https://clinicaltrials.gov/, 1 March 2022) there are two clinical studies (NCT00420134 and NCT02943889) in which MSCs are initially differentiated into HLCs and then transplanted to patients with liver diseases. In 2009, Kharaziha et al. isolated BM-MSCs from end-stage liver diseases, differentiated these cells into HLCs and then injected them into the portal veins of patients with LC [74]. To effectively induce hepatic differentiation, the cells were cultured in the presence of HGF, DEX and OSM and then injected through one of the main branches of the portal vein under ultrasound guidance [74]. According to their results, all patients showed objective improvements, with a reduced requirement for therapeutic laparocentesis and with a reduced MELD score and normal creatinine levels as well as improved liver function and an increase in serum albumin [74]. Therefore, it was suggested that injection with HLCs may improve clinical indices of liver function in end-stage liver diseases [74] (Table 1).

Liver stem cells present another potential candidate for liver regeneration. Fetal tissue contains a significant amount of stem and progenitor cells that may be valuable for therapeutic approaches. It has been reported that FLPCs and progenitor cells have also been used as a potential therapy for liver diseases [20]. According to the NCT01013194 clinical trial, the transplantation of 5–10 × 10^8^ FLPCs in patients with LC resulted in an innovative and safe treatment for patients with end-stage chronic liver failure [75] (Table 1).

## 6. Generation of Liver Organoids

Organoids represent a novel approach in the frame of liver regenerative medicine that could be utilized as transplantable units, disease models, toxicological studies or tools for drug discovery. Organoids, which are 3D self-organized scaffolds based on cellular structures that are derived from differentiated ESCs, iPSCs or adult stem cells (ASCs), are grown in a serum-free condition and display the full spectrum of cellular types in a tissue [20]. More specifically, a liver organoid is a 3D multicellular spherical structure made of one or more liver cell types [76]. Brassard et al. developed some parameters in order to control the process of building a 3D organoid [77]. Importantly, it has been suggested that the choice and control of the number of cells and the density of the dissociation–aggregation phase is mandatory, including the best culture conditions and the combination of growth factors to specifically guide the differentiation process. In liver organoids, the molecular signaling pathways that regulate liver embryonic development, such as HGF, FGF, BMP, Wnt and TGF, could guide the growth factor supplementation, which promotes hepatic progenitor migration, development and survival [78]. Furthermore, in order to build a 3D organoid, it is required to supply (if necessary) some pre-defined extrinsic forces to improve cell-to-cell and cell-to-extracellular-matrix (ECM) interactions [77]. Because the ECM plays a crucial role in supporting cell proliferation, improving cell adherence and the dispersal of nutrients and growth factors, stem/progenitor cells must be in a strict contact with its components, such as collagen, laminin and fibronectin [16]. Finally, micropatterning, microwells and microfluid dynamics are some of the designed geometries that can be utilized to support cell and organoid growth [77].

Liver organoid technology simulates the morphological and physiological properties and tissue-specific functions of the liver in a dish through the self-organization of cell populations that mimic the liver development process [79]. Liver organoids could be derived either from the culture of a single cell type, such as adult stem cells, LPCs or iPSCs, or from a multi-type cell co-culture, such as:A combination of iPSCs with human umbilical vein endothelial cells (HUVECs) and MSCs;Hepatocytes and stromal cells;Primary liver tumor cells or damaged liver cells [80].

The single type of cell culture ensures the proliferation and self-organization of a homogenous cell population and is easier to form, while co-culture of multiple cells types can better mimic the liver organ structure [34,38].

### 6.1. Single Cell-Type Culture for Organoid Development

The first single cell-type culture of a human ASC-derived organoid was developed by culturing LGR5^+^ biliary cells derived from a liver injury by adding Rspo1, EGF, FGF10, HGF, nicotinamide, cAMP agonist and TGF*β* receptor inhibitor to the differentiation medium [81]. Since then, other large-scale culturing methods for ASC-derived liver organoids have been established [82]. A number of studies have already proposed the development of single-cell culture-derived liver organoids by the use of liver progenitor cells. First, Hu et al. embedded human mature primary hepatocytes in Matrigel to form liver organoids using molecule inducers, such as Rspo1, EGF, FGF7, FGF10, HGF and TGF*β* inhibitor (Figure 1) [83]. As a result, they developed human fetal hepatocyte organoids (Hep-Orgs), which were cultured for more than 11 months [83]. After that, in 2020, Hendriks et al. established a protocol with similar culture conditions that facilitate the long-term expansion of human fetal hepatocytes as organoids. [84].

Furthermore, the use of iPSC-derived liver organoids has already been proposed for the treatment of liver diseases. Because liver progenitors arise from the endoderm, iPSCs must first be differentiated into definitive endoderm to mimic liver development [85] and further exhibit hepatic maturation in alternate normoxia and hypoxia conditions [86]. Following that, liver organoids will be formed over 2D monolayers of mature hepatocytes that will then be collected and embedded on Matrigel [86]. In 2022, Messina et al. generated human iPSC-derived hepatocytes (iHeps) that self-assembled as organoids (iHep-Orgs) [86]. In particular, human iPSCs were first differentiated into hepatoblasts (iHBs) using growth factors and cytokines according to other already published protocols [86,87]. Then, iHBs were developed into iHep-Orgs using HGF, vitamin K, OSM and Dex, and their culture extended until day 38, as they were still viable, with the absence of necrotic cores [86] (Figure 1). Their successful development of those iHep-Orgs was then confirmed by several studies, including gene expression for *EPCAM* and *CXCR4*, the absence of pluripotent markers and the presence of hepatoblast markers as well as by the well-defined morphology [20,80,84,88,89,90,91].

### 6.2. Multi-Type Cell Co-Culture for Organoid Development

To ensure cell-to-cell contacts during liver development, multi-cell co-cultures are able to generate a complex vascularized 3D structure in dishes [92]. In 2013, Takebe et al. generated, for the first time, a vascularized and functional human liver from human iPSCs by the transplantation of liver buds that were created in vitro [93]. More specifically, iPSCs were co-cultured with HUVECs and MSCs and spontaneously self-organized into macroscopically visible 3D cell aggregates showing endothelial network and the expression of hepatic-specific marker genes [93]. After that, the whole liver buds were connected to the recipient’s vasculature to generate functional vascular networks, promoting liver bud development [93]. Since MSCs can provide a number of signals promoting hepatocyte growth and development, co-culture of BM-MSCs and hepatocytes exhibited higher rates of hepatocyte-specific functions, maintained hepatocyte metabolism and generated higher quality hepatocyte organoids [94]. Although some studies suggested that MSC and HUVEC paracrine signals can both drive hepatocyte differentiation, both must co-exist to allow for cell-to-cell interaction and organization into a 3D liver organoid [95] (Figure 1). In 2018, Nie et al., successfully generated human liver organoids derived from multiple cells from a single donor, and these organoids were finally able to rescue a mouse model of ALF [96]. In more detail, umbilical cord endothelial cells (UC-ECs) were able to generate donor-derived human iPSCs, which then were able to efficiently differentiate into pure definitive endoderm and further into the hepatic lineage, and with a co-culture of these three cell types (UC-ECs, iPSCs and MSCs), the investigators successfully generated a single-donor cell-derived liver organoid [96]. In 2021, Qiu et al. generated a functional 3D sheet-like human hepatocellular carcinoma (HCC) organoid in vitro by co-culturing luciferase-expressing Huf7 cells, hiPSCs-derived endothelial cells and hiPSC-derived MSCs [97]. According to their results, once they added iPSC-MSCs, the organization of the 3D HCC organoid was easier, suggesting that iPSC-MSCs play a crucial role in promoting HCC organoid growth [97].

Although iPSCs appear to be beneficial in 3D liver organoid development, some studies tested a co-culture of MSCs and HUVECS with other cell types, such as amniotic stem cells (ASCs) (Figure 1), resulting in the generation of a 3D structure with polarity and hepatic-like glycogen storage [98]. Co-culturing methods of human hepatocytes, HUVECS and MSCs have also been applied [99]. Moreover, the privilege of co-culturing human hepatocytes, HUVECs and MSCs has been tested in fusing hundreds of liver-bud-like spheroids using a 3D bioprinter [100]. Currently, studies for organoid development focus on co-culturing mesenchymal and endothelial cells, as these cells are known for their ability to regulate liver progenitor cells’ fate and growth [101].

Furthermore, the co-culturing of primary liver tumor [102,103,104] or damaged liver cells [79,105] has also been studied. Patient-derived tumor organoids (PDO) are 3D cell culture models that closely mimic the form and function of tumor tissue, illustrate cell-to-cell and cell-to-matrix interactions and have similar pathophysiological characteristics to differentiated tumor tissue in vitro [92]. Primary liver tumor organoids can be derived from liver tumor specimens, preserving the histological architecture, gene expression and genomic landscape of the original tumor, providing a tool for biomarker identification and drug screening [103]. Liver organoids can also be derived from patient biopsies with alpha-1-antitrypsin (ATT) syndrome, which is a useful tool for modeling disease pathology [80]. In addition, in 2021, McCarron et al. received wedge biopsies from patients with nonalcoholic steatohepatitis (NASH) and developed hepatic organoids, suggesting that these organoids can be used for personalized disease modeling and drug development [105]. Consequently, liver organoids can be derived from either healthy or injured patient tissues. Genetic modifications of these organoids enable disease modeling, the clarification of molecular pathogenesis, patient-specific drug testing, personalized medicine and biomarker discovery, causing them to be an alternative model for biomedical research [106,107].

## 7. Clinical Applications of Liver Organoids

Although no clinical studies have proceeded to mainstream medical therapies, there has been considerable progress in cell-based treatments for liver disease over the past decade. For this reason, organoid application has opened the way to some new approaches in the field of liver diseases therapies and liver transplantation. Organoids appear to be a perfect tool for understanding the development of liver diseases and drug screening [16]. Hepatocytes in a 3D structure represent a novel tool for providing liver cell treatment.

Currently there are two clinical trials investigating the therapeutic potential of 3D liver organoids. In the first trial (NCT02718235), the investigators are trying to establish a novel prognostic tool, the Hepatocellular Carcinoma Immune Score (HCCIS), for risk stratification for patients with HCC, which can be widely used in clinical practice. In more detail, they aim to generate a database with HCC-related tumor organoids. In addition, they aim to test the effect of CD8^+^ IL-33^+^ effector-memory cells on the hepatocellular carcinoma tumor organoids to establish the respective HCCIS risk groups. Furthermore, the HCCIS appears to be a risk stratification tool that is independent from clinical or descriptive criteria. Further, investigators aim to show that the different HCCIS risk groups differ not only in terms of immunological infiltration but also in terms of tumor biology. Accordingly, three HCCIS risk points, low, medium and high, with scores of 2, 1 and 0, respectively, were established.

The other clinical trial (NCT03896958) is about the PIONEER Initiative. PIONEER concerns the precision insights on N-of-1 ex vivo effectiveness research, providing access to functional precision medicine to any cancer patient with any tumor at any medical facility. In more detail, the tumor tissue, derived by biopsy or surgery, is kept in a variety of ways, including fresh and cryopreserved as a living biospecimen, and can be sent wherever the patient and the clinical team require further analysis. The use of biospecimens can make clinical studies more accessible, and its overall purpose is to enable best-in-class functional precision testing of patient tumor tissues to aid in the selection of the most appropriate treatment. Currently, this type of analysis includes organoid drug screening techniques. More specific data and samples will be utilized to improve current treatments and to better understand the changes that occur through the transition from disease to recovery at the molecular level. The clinical trials that use organoids in liver diseases are presented in Table 2.

## 8. Future Liver Therapy Perspectives

So far, MSC- and LPC-based clinical trials have shown promising results in liver amelioration, allowing extended clinical research in stem cell therapy. Several pre-clinical studies using not only MSCs but also EVs or secretomes derived from MSCs have presented important findings in therapy of liver fibrosis, LC, AHF, NASH and HCC, with a potential clinical “cell-free” usage [7,8,12]. The secretome can overcome the limitations of MSC transplantation and, in conjunction with the biotechnological applications that arise, is able to attenuate MSC effectiveness [108]. EVs can be engineered to target specific cells or to transfer the desired proteins and/or RNAs in the site of tissue injury [14].

Meanwhile, HLCs have also been reported to exert optimal characteristics for therapy and drug screening for liver diseases. In particular, the microencapsulation of MSCs or hepatocytes in biocompatible semipermeable alginate-based hydrogels has also been developed for the treatment of acute or chronic diseases, protecting cells from the immune system, antibodies and immune cells [109]. However, the maturation of HLCs remains a challenging procedure that is subjected to alternative approaches, such as growth factors and cytokines, gene manipulation including the CRISPR technique and 3D culture formation. The combination of engineered HLCs with the technologies of organoids and organs-on-a-chip may lead to a more functional formation of hepatic tissue and can help in gaining a deeper understanding of liver disease pathology [110].

## 9. Discussion

Clinical studies are focused on the usage of MSCs in liver diseases, including mostly LC and LF disorders, with promising results in liver pathology. It has been mentioned that the transplantation of MSCs derived from various sources, especially from BM and UC, ameliorated the liver disease symptoms, transaminase and metabolic marker levels, as well as the MELD and Child Pugh scores, allowing the further clinical investigation of stem-cell-based liver therapies. MSCs exhibit no side effects or any other immune-activation effect and can effectively migrate to the area of injured tissue. However, many considerations related to poor cell engraftment and the limited improvement of liver histology as well as to the low therapeutic impact among different groups [106,107] raise questions about the efficacy of MSCs in clinical applications and necessitate alternative solutions in liver therapy.

Interestingly, a recent clinical trial confirmed the therapeutic effect of MSC-derived HLC transplantation in patients with LC, opening the perspective for novel technologies in HLC generation from various MSC sources. Moreover, the transplantation of another type of progenitor cells, the LPCs, has been clinically tested in a well-tolerated treatment but without any representative therapeutic outcome, confirming the need for advanced research in the area of stem cell therapies.

Recently, the understanding of the molecular and genetic background of liver tissue as well as of liver diseases contributes to the design of more specific and successful liver therapies. Therefore, liver organoids and the organoid co-cultures with several cell types, including MSCs, are being developed in order to mimic the liver environment and patient disease phenotypes for liver therapy and drug screening. So far, few clinical attempts have been conducted, mostly for HCC patients, with the development of promising prognostic tools, such as HCCIS and PIONEER, for tumor biology. Given their promising applications, the ultimate goal is to perform tissue-specific organoid transplantation to restore or improve liver function and enhance the quality of life of patients.

The clinical approach for liver failure based on the transplantation of hepatocytes derived from MSCs and in conjunction with liver organoids can be considered as the most suitable therapeutic option for liver disease therapy [74,107,108,110]. Although, many limitations concerning the long-term efficacy, the dose and the route of administration of MSCs as well as the MSC modification for functionally generated HLCs and the appropriate type of liver disease for transplantation still need to be addressed in order to acquire more accurate treatments for patients with liver malfunctions.

## 10. Conclusions

According to the outcomes from recent clinical studies, MSCs, HLCs and LPCs are valuable tools in liver regeneration therapeutic approaches. Although there are important scientific challenges that must be addressed in order to develop safe and effective therapeutic modalities, such procedures may pave the way for stem-cell-based clinical practices and improve the quality of life of patients with liver disorders. In addition, basic research for disease modeling and drug development based on liver organoids represents a promising tool for biomedical applications in the field of liver diseases.

## Figures and Tables

**Figure 1 cells-11-01410-f001:**
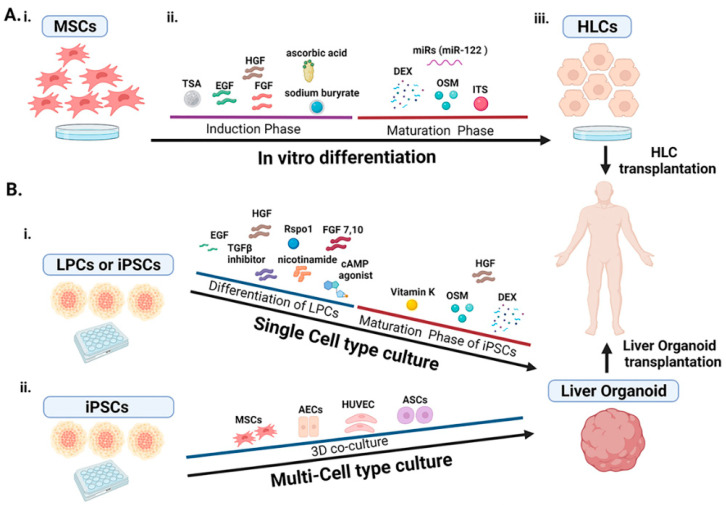
Generation of MSC-derived HLCs and liver organoids from iPSCs or LPCs for transplantation in patients with liver diseases. (**A**). In vitro (**i**) culture and (**ii**) differentiation of MSC into HPLCs. The differentiation process is divided into the induction and maturation phases with the addition of growth factors, molecules or miRs. (**iii**) The derived HLCs can be used for transplantation. (**B**). Generation of liver organoids from different cell types. (**i**) Single cell-type differentiation of LPCs in presence of growth factors and molecules and maturation of iPSCs in presence of several compounds that lead to liver organoid formation. (**ii**) Multi-cell-type culture of iPSCs with other cell types for liver organoid formation. The 3D co-culture of iPSCs with a variety of cell types enhances the liver organoid formation for transplantation in patients with liver diseases. MSCs, mesenchymal stromal cells; HLCs, hepatocyte-like cells; TSA, trichostatin; EGF, epidermal growth factor; HGF, hepatic growth factor; miR; micro-RNAs; DEX; dexamethasone; OSM, oncostatin M; ITS, insulin-transferin-sodium selenite; LPCs, liver progenitor cells; TGF*β*, transforming growth factor *β*; Rspo1, R-spondin 1; FGF, fibroblast growth factor; cAMP, cyclic adenosine monophosphate; AECs, amniotic epithelial cells; HUVEC, human umbilical vein endothelial cells; ASCs, amniotic stem cells. (Created by Biorender.com, 30 March 2022).

**Figure 2 cells-11-01410-f002:**
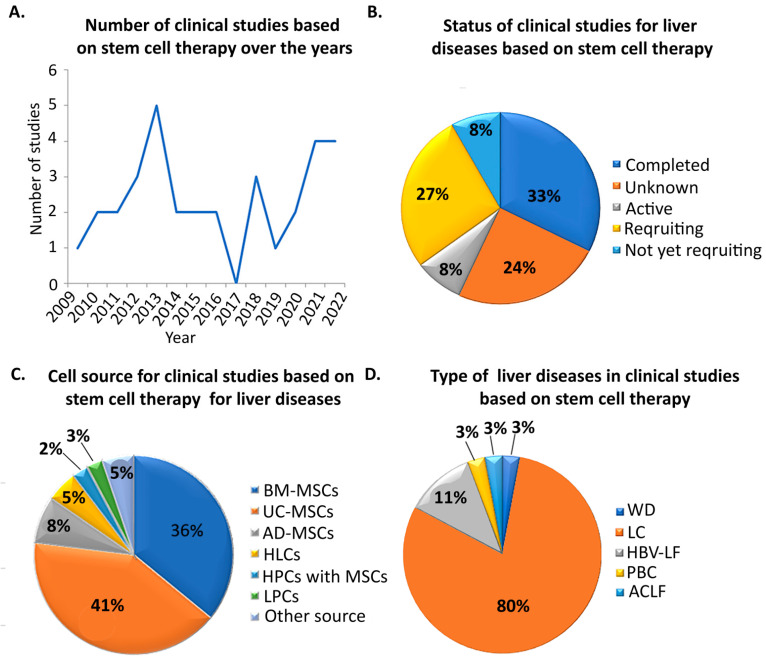
Summary of the clinical trials using MSCs, HLCs or LPCs for therapy of liver diseases. (**A**). The graph depicts the number of stem-cell-based studies that have been conducted over the years. (**B**). Pie chart that represents the current status of clinical studies for liver diseases based on stem cell therapy. (**C**). Pie chart that depicts the percentages of the different stem cell sources that have been used in clinical liver therapy. (**D**). Pie chart that represents the percentages of clinical studies for liver diseases based on stem cell therapies. BM-MSCs: Bone Marrow MSCs; UC-MSCs: Umbilical cord MSCs; AD-MSCs: Adipose Tissue MSCs; HLCs: Hepatocyte-Like Cells; HPCs: Hematopoietic Progenitor Cells; LPCs: human Liver Progenitor Cells; WD: Wilson’s Disease; LC: Liver Cirrhosis; LF: Liver Failure; PBC: Primary Biliary Cirrhosis; ACLF: Acute-on-Chronic Liver Failure.

**Table 1 cells-11-01410-t001:** Clinical Trials using MSCs and LPCs in Liver Diseases.

Title	NCT Number	Year	Country	Status	Cell Source	Disease	Outcome	Bibliography
Efficacy of Invitro Expanded BM Derived Allogeneic MSC Transplantation via Portal Vein or Hepatic Artery or Peripheral Vein in Patients With Wilson Cirrhosis	NCT01378182	2014	Turkey	completed	BM-MSCs	WD	(NS)	(NI)
Dose Finding Study to Assess Safety and Efficacy of Stem Cells in LC	NCT01591200	2016	India	completed	BM-MSCs	LC	Reasonably safe dose of up to 2.5 million cells/kg body weight	[63]
REVIVE (Randomized Exploratory Clinical Trial to Evaluate the Safety and Effectiveness of Stem Cell Product in Alcoholic LC Patient)	NCT01875081	2016	Korea	completed	BM-MSCs	LC	Reduction of collagen deposition	[64]
The Effectiveness and Safety for MSCs for Alcoholic LC	NCT01741090	2012	Korea	unknown	BM-MSCs	LC	Improved histology, Child Pugh’s Score	[65]
Study to Evaluate Hepatic Artery Injection of Autologous hBM-MSCs in Patients With Alcoholic LC	NCT03838250	2020	United States	recruiting	BM-MSCs	LC	(NS)	(NI)
Clinical Trial to Evaluate the Efficacy and Safety of Cellgram-LC Administration in Patients With Alcoholic Cirrhosis (Cellgram-LC)	NCT04689152	2021	Korea	recruiting	BM-MSCs	LC	(NS)	(NI)
BM-MSC Transplantation in LC Via Portal Vein	NCT00993941	2010	China	active, not recruiting	BM-MSCs	LC	(NS)	(NI)
Autologous MSC Transplantation in LC	NCT01499459	2012	Turkey	active, not recruiting	BM-MSCs	LC	Improvement in Alb and MELD scores	[66]
MSC Transplantation in Decompensated Cirrhosis	NCT00476060	2011	Iran	active, not recruiting	BM-MSCs	LC	No beneficial effect.	[67]
Transplantation of Autologous MSCs in Decompensate Cirrhotic Patients With Pioglitazone	NCT01454336	2014	Iran	completed	BM-MSCs	LC	(NS)	(NI)
Therapeutic Effects of Liver Failure Patients Caused by Chronic Hepatitis B After Autologous MSC Transplantation	NCT00956891	2010	China	completed	BM-MSCs	HBV-related LF	No markedly improved long-term outcomes	[68]
Safety and Efficacy of Human BM-MSCs for Treatment of HBV-related Liver Cirrhosis	NCT01724697	2012	China	unknown	BM-MSCs	LC	(NS)	(NI)
Trial of MSC Transplantation in Decompensated LC	NCT03209986	2018	China	unknown	MSCs	LC	(NS)	(NI)
Human UC-MSC Transplantation for Patients With Decompensated Liver Cirrhosis	NCT01342250	2011	China	completed	UC-MSCs	LC	(NS)	(NI)
UC-MSCs for Patients With Liver Cirrhosis	NCT01220492	2018	China	completed	UC-MSCs	LC	Improved liver function	[69]
UC-MSCs for Decompensated Cirrhosis (MSC-DLC-1)	NCT05227846	2022	China	not yet recruiting	UC-MSCs	LC	(NS)	(NI)
MSC treatment for Decompensated LC	NCT03945487	2019	China	recruiting	UC-MSCs	LC	(NS)	(NI)
Treatment with UC-MSCs for Decompensated Cirrhosis	NCT05121870	2021	China	recruiting	UC-MSCs	LC	(NS)	(NI)
UC-MSCs for Decompensated Cirrhosis (MSC-DLC-2)	NCT05224960	2022	China	not yet recruiting	UC-MSCs	LC	(NS)	(NI)
UC -MSC Transplantation for Children Suffering From Biliary Atresia (UCMSCBA)	NCT04522869	2020	Vietnam	recruiting	UC-MSCs	LC	(NS)	(NI)
Study of Decompensated Alcoholic Cirrhosis Treatment by Stem Cells	NCT05155657	2022	China	recruiting	UC-MSCs	LC	(NS)	(NI)
UC-MSC Transplantation Combined With Plasma Exchange for Patients With Liver Failure	NCT01724398	2013	China	unknown	UC-MSCs combined with PE	HBV-related LF	Decreased levels of total bilirubin, alanine aminotransferase, aspartate transaminase and MELD score	[70]
Safety and Efficacy of UC-MSCs for Treatment of HBV-related Liver Cirrhosis	NCT01728727	2013	China	unknown	UC-MSCs	LC	(NS)	(NI)
Safety and Efficacy of hMSCs for Treatment of Liver Failure	NCT01218464	2013	China	unknown	UC-MSCs	HBV-related LF	(NS)	(NI)
UC-MSCs for Patients With Primary Biliary Cirrhosis	NCT01662973	2013	China	recruiting	UC-MSCs	PBC	1. Improvement in Alb, T-BIL and MELD score. 2. Decrease in serum alkaline phosphate andg-glutamilitransferase levels.	[71]
Safety of UC-MSC Transfusion for ACLF Patients	NCT04822922	2021	China	not yet recruiting	hUC-MSCs	ACLF	(NS)	(NI)
UC-MSCs (19#iSCLife^®^-LC) in the Treatment of Decompensated Hepatitis b Cirrhosishepatitis b Cirrhosis	NCT03826433	2022	China	recruiting	UC-MSCs	LC	(NS)	(NI)
UC-MSC Transplantation for Decompensated Hepatitis B Cirrhosis	NCT05106972	2021	China	recruiting	UC-MSCs	LC	(NS)	(NI)
Combination of Autologous MSC and HSC Infusion in Patients with Decompensated Cirrhosis	NCT04243681	2020	India	completed	HPCs (CD34+) combined with MSCs	LC	Improvement in the MELD score and in serum albumin levels	[72]
A Clinical Study to Evaluate the Safety and Efficacy of MSCs in Liver Cirrhosis	NCT01877759	2013	India	unknown	BM-MSCs and UC-MSCs	LC	(NS)	(NI)
Safety and Efficacy of Diverse MSCs Transplantation for Liver Failure	NCT01844063	2013	China	unknown	BM-MSCs and UC-MSCs	HBV-related LF	(NS)	(NI)
Liver Regeneration Therapy by Intrahepatic Arterial Administration of Autologous Adipose Tissue Derived Stromal Cells	NCT01062750	2015	Japan	completed	AD-MSCs	LC	(NS)	(NI)
Clinical Trial Study About Human Adipose-Derived Stem Cells in the LC	NCT02297867	2018	Taiwan	completed	AD-SCs	LC	Repaired liver fibrosis	[73]
A Study of ADR-001 in Patients With LC	NCT03254758	2021	Japan	recruiting	AD-MSCs	LC	(NS)	(NI)
Improvement of Liver Function in LC Patients After Autologous MSC Injection: Phase I-II Clinical Trial	NCT00420134	2009	Iran	completed	MSC-derived HLCs	LC	Improved liver function	[74]
Stem Cell Transplantation in Cirrhotic Patients	NCT02943889	2016	Egypt	unknown	MSC-derived HLCs	LC	(NS)	(NI)
Human Fetal Liver Cell Transplantation for Treatment of Chronic Liver Failure	NCT01013194	2015	Italy	completed	LPCs	LC	(NS)	[75]

(NS) Non-Specified, (NI) Non-Identified. BM-MSCs: Bone Marrow MSCs; UC-MSCs: Umbilical cord MSCs; AD-MSCs: Adipose Tissue MSCs; HLCs: Hepatocyte-Like Cells; HPCs: Hematopoietic Progenitor Cells; LPCs: human Liver Progenitor Cells; T-BIL: PE, Total Bilirubin; PE: Plasma Exchange; WD: Wilson’s Disease; LC: Liver Cirrhosis; LF: Liver Failure; PBC: Primary Biliary Cirrhosis; ACLF: Acute-on-Chronic Liver Failure.

**Table 2 cells-11-01410-t002:** Clinical Trials Based on the Use of Organoids for Liver Diseases.

Title	NCT Number	Year	Country	Status	Cell Source	Disease	Outcome	Bibliography
Prospective, Multicenter HCCIS Evaluation Study (HCCIS)	NCT02718235	2016	Germany	unknown	(NS)	HCC	(NS)	(NI)
The PIONEER Initiative: Precision Insights On N-of-1 Ex Vivo Effectiveness Research Based on Individual Tumor Ownership (Precision Oncology) (PIONEER)	NCT03896958	2020	United States	Recruiting	(NS)	HCC	(NS)	(NI)

(NS) Non-Specified, (NI) Non-Identified. HCC: Hepatocellular Carcinoma.

## Data Availability

Not Applicable.

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
