# Peer review of "The Potential Clinical Use of Stem/Progenitor Cells and Organoids in Liver Diseases"

_cells, 2022, doi:10.3390/cells11091410_

Round 1
Reviewer 1 Report
Readers are provided with a comprehensive review on the topic of cellular therapy for liver diseases. The literature review is written in great detail, the most up-to-date research on this topic is presented, general tables and illustrations are presented, which facilitate the analysis of the data presented. The only comment about the title. It seems to me that the literature review addresses a wider range of issues than the involvement of MSCs and the cells and organelles derived from them in the treatment of liver disease.
Author Response
We thank this and the other reviewers for their comments and for finding our review interesting. The changes suggested have greatly improved the manuscript. We have therefore taken on board the concerns raised and have addressed specific points below in red.
We agree with the reviewer’s comment regarding the title. Thus, we have now revised the title accordingly to: “The potential clinical use of stem/progenitor cells and organoids in Liver Diseases”.

Reviewer 2 Report
The authors' review manuscript discussed that the characteristics of stem cell-based therapies for liver disease, and address to present the current status and the future perspective of using hepatocyte-like cells, liver progenitor cells or liver organoid in clinical trials. The review manuscript did well organized in this field and addressed to our present limit to liver disease treatment and cell therapy and problem, and show to future study.
Author Response
We thank the reviewer for finding our review interesting and well organized.
Reviewer 3 Report
The content of this review should be shortened to focus more on MSC-derived HLC. In particular, MSC research progress in clinical trials should be highlighted, because MSC-derived HLC is already at a disadvantage in basic research compared to cell sources such as HPC and iPSC.
Author Response
We would like to thank the reviewer for this suggestion. Accordingly, we have shortened the HLC section as requested (Please see section 3.HLC generation from MSCs) and made the appropriate changes. Further, we have proof read the revised paper, and have corrected the English and Typing errors as specified above in response to Reviewer 3.
Reviewer 4 Report
1.The content of this manuscript is inconsistent with the title.The tiltle is “The clinical impact of MSC derived Hepatocyte like cells and organoids in Liver Diseases”. However, other stem cells such as iPSCs , ESCs and LPCs and et al were all mentioned a lot not only in the section of abstract, but also in main text. I would advise authors to focus only on MSCs or MSC-derived HLCs in the application of liver diseases. And it is better to discuss the use of MSCs (not differentiated in vitro) and MSCs-derived HLCs separately, as their mechanisms of action are quite different in many ways.
2. Abbreviation (MSC) in title is not recommended. Line 231 and line 342, use abbreviations (MSCs and HLCs ) instead of the full names.
3. Ref 11 is the same as Ref 13.
Author Response
1.The content of this manuscript is inconsistent with the title. The tiltle is “The clinical impact of MSC derived Hepatocyte like cells and organoids in Liver Diseases”. However, other stem cells such as iPSCs , ESCs and LPCs and et al were all mentioned a lot not only in the section of abstract, but also in main text. I would advise authors to focus only on MSCs or MSC-derived HLCs in the application of liver diseases. And it is better to discuss the use of MSCs (not differentiated in vitro) and MSCs-derived HLCs separately,as their mechanisms of action are quite different in many ways.
We would like to thank the reviewer for this suggestion. We agree with the reviewer’s comment regarding the title. Thus, we have now revised the title accordingly to: “The potential clinical use of stem/progenitor cells and organoids in Liver Diseases”. In addition, we have revised the respective section for HLCs and MSC as suggested.
- Abbreviation (MSC) in title is not recommended. Line 231 and line 342, use abbreviations (MSCs and HLCs ) instead of the full names.
We have corrected the text as suggested.
- Ref 11 is the same as Ref 13.
We have revised the reference section and removed the duplicated references. In addition, we have proof read the revised paper, and have corrected the English and typing errors in response to Reviewer 4.

Round 2
Reviewer 4 Report
Accept